# Comparing Deep Learning and Statistical Methods in Forecasting Crowd Distribution from Aggregated Mobile Phone Data

**Alket Cecaj** [1,2,*] , **Marco Lippi** [1,2] , **Marco Mamei** [1,2] and **Franco Zambonelli** [1,2]

1    Dipartimento di Scienze e Metodi dell'Ingegneria, University of Modena and Reggio Emilia, 42122 Reggio Emilia, Italy; marco.lippi@unimore.it (M.L.); marco.mamei@unimore.it (M.M.); franco.zambonelli@unimore.it (F.Z.)
2    Artificial Intelligence Research and Innovation Center (AIRI), University of Modena and Reggio Emilia, 41125 Modena, Italy
*    Correspondence: alket.cecaj@unimore.it

**Abstract:** Accurately forecasting how crowds of people are distributed in urban areas during daily activities is of key importance for the smart city vision and related applications. In this work we forecast the crowd density and distribution in an urban area by analyzing an aggregated mobile phone dataset. By comparing the forecasting performance of statistical and deep learning methods on the aggregated mobile data we show that each class of methods has its advantages and disadvantages depending on the forecasting scenario. However, for our time-series forecasting problem, deep learning methods are preferable when it comes to simplicity and immediacy of use, since they do not require a time-consuming model selection for each different cell. Deep learning approaches are also appropriate when aiming to reduce the maximum forecasting error. Statistical methods instead show their superiority in providing more precise forecasting results, but they require data domain knowledge and computationally expensive techniques in order to select the best parameters.

**Keywords:** forecasting; time series; crowd distribution; aggregated mobile phone data; deep neural networks

## 1. Introduction

Being able to model and eventually forecast population density and crowd distributions in a smart city scenario is mainly about studying and analyzing data in space and time domain. A very useful kind of data able to describe human presence and mobility in urban and rural areas is mobile phone data. There are different scientific works that have been studying mobile phone data in depth thus revealing the different patterns that emerge [1–4]. Another line of research for aggregated mobile phone data such as in [5] deals with modelling them as a time series, so as to explore their predictability by using different forecasting methods. Although a large part of predictive analytics is based on modelling and developing forecasting algorithms, there is still a lack of understanding of which forecasting method is best for which time series forecasting scenario. The forecasting methods that we use in this study belong to two broad categories: statistical approaches and deep learning approaches. As we show in this work, both categories have advantages and disadvantages, thus the purpose of our study is not to establish an absolute winner between the two. Instead, we aim to underline the strengths and weaknesses of each category of methods and, most importantly, to investigate which one might be the most appropriate for a specific forecasting scenario.

Following this direction, the main contribution of our work is that of evaluating the forecasting performance of different statistical and deep learning methods in predicting mobile activity and thus

the crowd distribution represented by such activity in aggregated mobile data. More in detail in Section 2, we present the aggregated dataset, its extension in space and time as well as its main features. In Section 3 we present the statistical and deep learning methods that we use for performing forecast. In the same section we illustrate how we proceed with model evaluation and the error forecasting measures that we use. In Section 4 we illustrate three different forecasting scenario and the experiments we perform in order to explore such scenario. In Section 5 we go through the related work on relevant forecasting methods and finally in Section 6 we present our conclusions.

## 2. Dataset

As we stated above, mobile phone can give researchers a great proxy for studying human activity, to better model it, describe and eventually predict it. However, different studies have shown that these data can harm an individual's privacy [6]. To comply with the European regulation on data protection (GDPR), (General Data Protection Regulation (EU) 2016/679 (GDPR))—the EU regulation law on data protection and privacy) the mobile phone carriers release the data only after having aggregated the mobile activity. Aggregated data counts the number of mobile phones in a given area of the territory over a given time period without any reference to the individuals being monitored (i.e., counted)—see Figure 1.

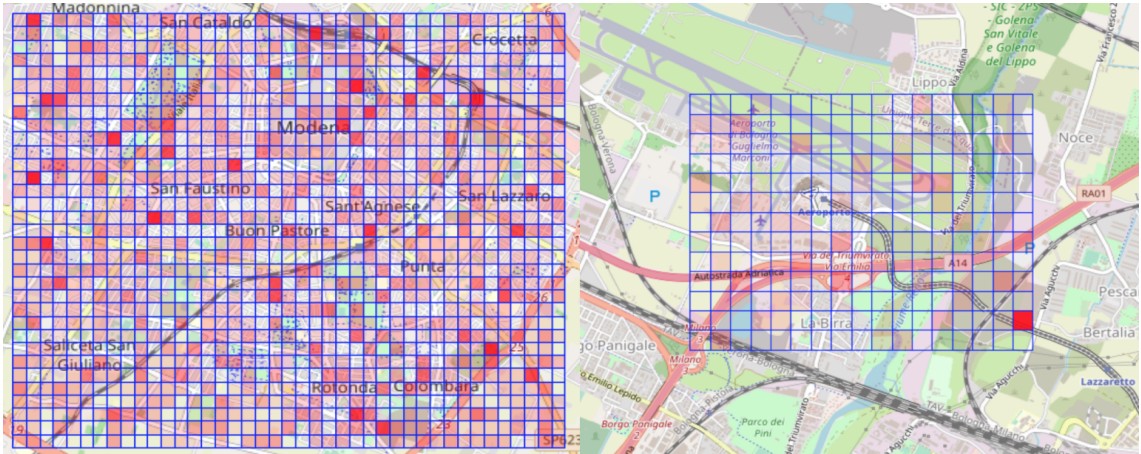

**Figure 1.** Grid of aggregated CDR mobile data. The grid on the left is 5 km$^2$ and covers a medium size city center. The grid on the right is smaller 2.5 km$^2$ and covers an airport.

### 2.1. Dataset as a Grid over the Urban Area

In particular, our dataset consists of two grids that cover two urban areas of 5 and 2.5 km$^2$, respectively. Figure 1 shows the two grids where the color of each cell is based on the number of mobile phones (approximately) located inside the given cell in that particular moment. The interval of sampling is fixed to every 15 min, which is a time granularity that allows for very precise reconstruction of the underlying mobile activity i.e., the number of mobile phones. Each cell in both grids covers an area of 150 m$^2$, and the observations in our dataset cover a time span of four months from mid April to mid August 2017. As illustrated in Figure 2, each cell of the grid contains the number of mobile phones as a function of time. Once modeled as time series, the aggregated mobile data can be used to forecast the underlying human presence and activity.

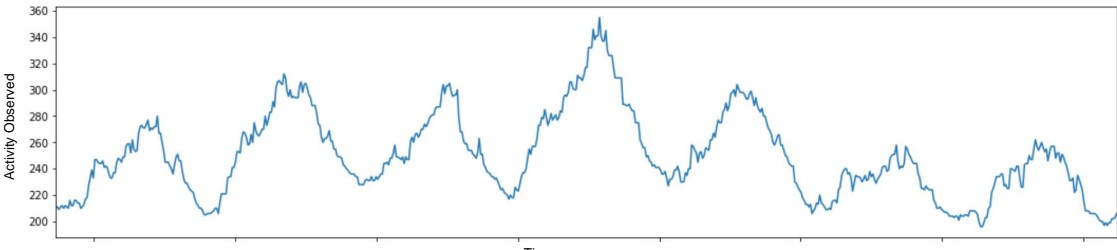

**Figure 2.** Number of mobile phones observed in a given cell during a seven days period. The number of mobile phones present in the cell appears to be periodic in time. It peaks between midday and the afternoon and reaches the lowest point between the midnight and the morning.

*2.2. Dataset as Time Series of People Presence*

Although the general format of the dataset is that of a large grid, each cell of the grid itself acts as a sensor of the underlying human activity. This is because each cell contains the observations in time of the number of mobile phones in the corresponding piece of land that the cell itself covers. These pieces of land may contain different types of infrastructure such as roads, parks, residential or industrial buildings, airports or stadiums. As a consequence, the mobile activity in every cell may result in very different levels of observation and patterns. There are however general patterns that occur in the majority of urban cells (i.e., excluding purely residential and leisure areas) such as that of having a peak activity between midday and afternoon or a pattern due to activity during business days and weekend days such as in Figures 2 and 3, respectively.

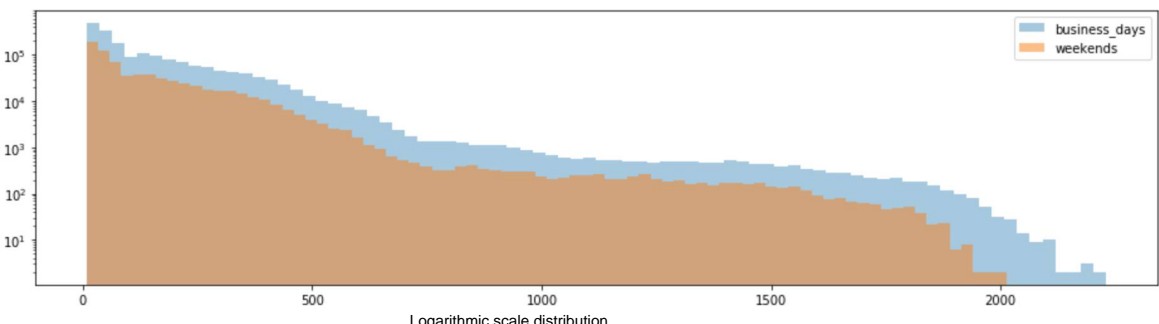

**Figure 3.** The histogram graphics in logarithmic scale shows the distribution of the number of mobile phones in all the cells of the largest one of the two grids. In particular the distribution in light blue regards the observations during business days while the brown one is about the observations seen during the weekends.

## 3. Methodology

In this section, we explain the methodology we adopt to compare the forecasting performance between deep learning and statistical approaches. To make the process clear and as uniform as possible we have formulated a series of steps that go from discovering relationships in the data, and through data transformation, to model the definition and the evaluation of forecasting performance measures. We start by investigating the possible relationship that exist between observations in time. For example, knowing how strongly correlated are observations in time $t$ with those in the successive time steps is of two-fold importance. On the one hand, it will determine the data transformations for our forecasting models. On the other, it will influence the choice itself of the statistical and deep learning models that we are experimenting with. Finally, the forecasting performance measures are a direct consequence of this investigation work as we will explain in the following sections.

### 3.1. Auto-Correlation in the Time-Series

As explained earlier, assessing the degree of auto-correlation of the time series in our data set is a key task in forecasting. As data in a time series is a sequence of values, it would be interesting to know how much does the observation at time $t$ depend on the one at time $t + 1$.

With a very simple experiment we scatter plot the observation $y$ at time $t$ and at time $t + 1$ in a cumulative way for all the time series in the grid. The plot is shown in Figure 4, and it shows a strong linear correlation along the main diagonal, but there is also a weak one below the main diagonal and a non-linear one above it. These correlations in the data are important for a forecasting model. The more the model will be able to capture such linear and non linear relationships the more it will be successful on forecasting them.

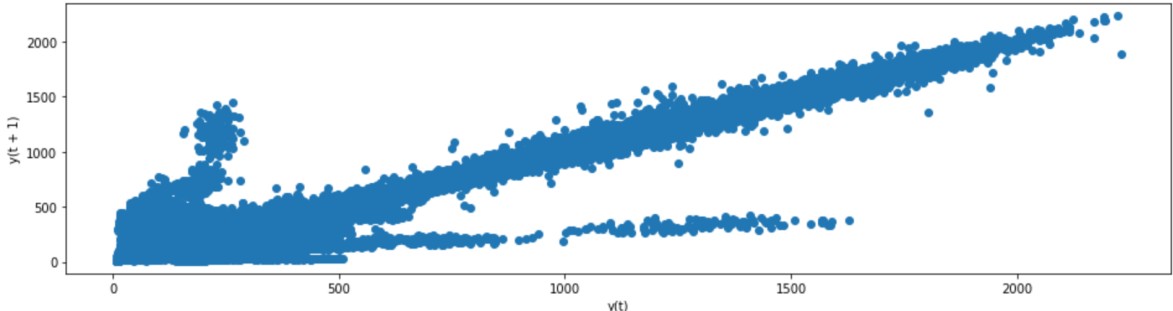

**Figure 4.** The scatter plot above shows the relationship between observation $y$ at time $t$ and that at time $t + 1$ for all the cells in the grid. The relationship is mainly linear, but non-linear behaviors can also be noticed.

Following this approach we correlate the observations at time $t$ with those at time $t + 1$, $t + 2$, ..., $t + n$, and we plot this auto-correlation as in Figure 5, which shows a strong correlation up to 15–18 steps. This means that lagged data can indeed be used as features to perform forecasting. However, other than linear relationships, there are also non-linear dependencies between the observed variable and its lags. For example, as mentioned above, the auto-correlation scatter plot in Figure 4 shows that besides the main diagonal there are two other directions along which the observed variable at time $t$ correlates with its lagged value at $t + 1$.

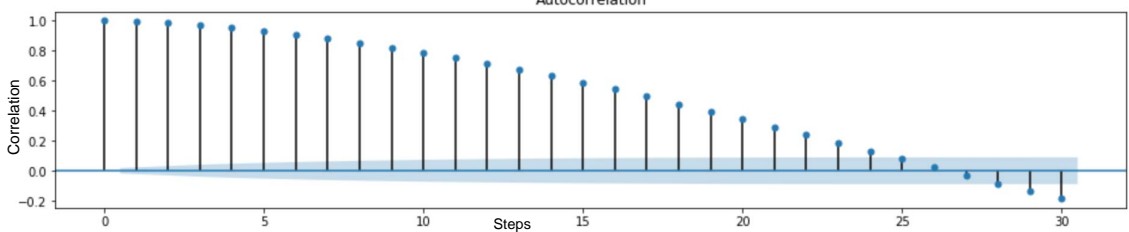

**Figure 5.** Graphics showing for every time step the degree of auto-correlation in the data from $t_0$ to $t30$ which appear to be strong up to 18 steps. The gray area shows the uncertainty on the degree of auto-correlation, which remains low.

### 3.2. Data Transformation

Another important point that we want to highlight in this section is that of data transformation and in particular data decomposition. Through experimentation we have found that by using STL decomposition [7] on the time series, we can improve our forecasting performance [8].

STL decomposition can handle any type of seasonality and has the nice property of transforming data so that forecasting approaches are more robust to outliers. More precisely, if the time series contains unusual observations such as spikes and anomalies, forecasting results will not be severely

affected. The process of STL decomposition is shown in Figure 6, where each observation $y_t$ is decomposed in trend ($T_t$), seasonal ($S_t$) and residual ($I_t$) components:

$$y_t = T_t + S_t + I_t \tag{1}$$

We use the decomposition process to perform forecasting on each component separately, and then sum up the three forecast values in order to compute the final prediction as in the following equation.

$$\widehat{y} = \widehat{T_t} + \widehat{S_t} + \widehat{I_t} \tag{2}$$

Another kind of data transformation we applied is differentiation, in order to make the time series stationary. The process simply consists of subtracting the observation at time $t + 1$ from that at time $t$. Thus, a differentiation-based data transformation is needed in order to apply statistical forecasting methods such as ARIMA for example.

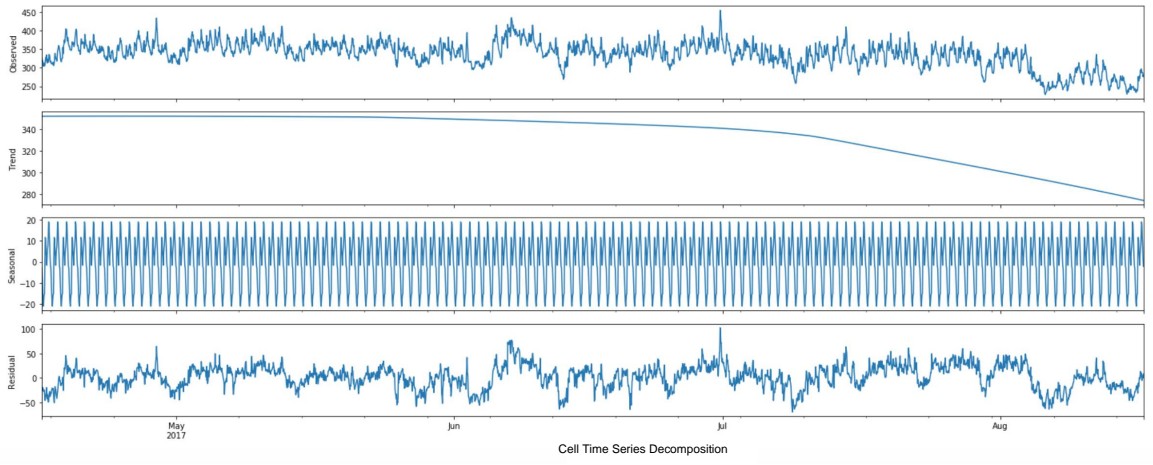

**Figure 6.** This figure shows the STL decomposition of the time series of one cell in the grid. Each time-series of a cell can be expressed as the sum of its components: trend, seasonal and residual.

Moreover, in order to apply deep neural networks forecasting methods on our uni-variate time series data, we had to transform it into a supervised learning dataset, by considering a rolling window and a number $n$ of lagged variables. Figure 7 shows an example, where $S$ is the uni-variate time series, while $X$ and $y$ are obtained by lagging $S$ for a given number of steps. From a learning perspective, $X$ is the set of features and $y$ contains the labels (supervisions), in this case for a 6-steps-ahead forecasting scenario.

```
  S                               X                                                   y
238.0 [238. 232. 231. 230. 233. 233. 230. 226. 224. 220. 218. 213.] [213. 206. 204. 203. 203. 202.]
232.0 [232. 231. 230. 233. 233. 230. 226. 224. 220. 218. 213. 213.] [206. 204. 203. 203. 202. 198.]
231.0 [231. 230. 233. 233. 230. 226. 224. 220. 218. 213. 213. 206.] [204. 203. 203. 202. 198. 198.]
230.0 [230. 233. 233. 230. 226. 224. 220. 218. 213. 213. 206. 204.] [203. 203. 202. 198. 198. 197.]
233.0 [233. 233. 230. 226. 224. 220. 218. 213. 213. 206. 204. 203.] [203. 202. 198. 198. 197. 194.]
233.0 [233. 230. 226. 224. 220. 218. 213. 213. 206. 204. 203. 203.] [202. 198. 198. 197. 194. 197.]
230.0 [230. 226. 224. 220. 218. 213. 213. 206. 204. 203. 203. 202.] [198. 198. 197. 194. 197. 196.]
226.0 [226. 224. 220. 218. 213. 213. 206. 204. 203. 203. 202. 198.] [198. 197. 194. 197. 196. 196.]
```

**Figure 7.** An example of the data after converting the uni-variate time series of a single cell $S$ into a supervised dataset made up of pairs $(x, y)$, by using lags of the observed variable $X$, and a rolling window through the whole time series.

### 3.3. Statistical Methods

In this subsection, we explain in detail the characteristics of each forecasting method we apply, as well as the reasons for considering each one of them. We describe first the statistical methods which

have historically dominated the time series forecasting research landscape. The main reason we have chosen to experiment with statistical methods is their ability to capture the linear relationships that exist in the temporal structure of our time series data as shown in Figure 4. Particularly good in this task are auto-regression models and among the many ARIMA forecasting method.

### 3.3.1. Seasonal Average

We started with a baseline, naïve forecasting method which provides a point of comparison for all the other forecasting techniques that will follow. We define this forecasting method simply as the seasonal average of the signal, i.e., the mean of all observations collected in the same day of the week and in the same time of the day. For example, when predicting the activity levels of a cell on a given Monday at 9:00 o'clock AM, we compute the mean of all the observations collected on every Monday at 9:00 AM inside the training set.

### 3.3.2. Autoregressive Models

Autoregression makes the assumption that the observations at the current and previous time steps are useful to predict the value at the next time step, by exploiting auto-correlation in the time series. The stronger the correlation between the output variable and a specific lagged variable, the more impact that input variable will have on the final prediction. This process is also called serial correlation because of the sequential nature of time series data. Although simple, this model can give accurate forecasts on a range of problems, and this is the main reason we considered it for this analysis.

### 3.3.3. The Arima Model

ARIMA is one of the most popular and widely used statistical methods for time series forecasting. The acronym stands for AutoRegressive Integrated Moving Average. The model is defined using three distinct input parameters:

- $a$, which is the lag order or the number of lag observations included in the model;
- $d$, which is the number of times that the raw observations are differentiated;
- $q$, which is the size of the moving average window used to compute the mean.

So for example if $a = 0$, $d = 2$ and $q = 2$ the forecast value $\widehat{Y_t}$ that ARIMA will predict will be equal to :

$$\widehat{Y_t} = 2Y_{t-1} - Y_{t-2} - \theta_1 e_{t-1} - \theta_2 e_{t-2}. \tag{3}$$

where $\theta_1$ and $\theta_2$ are the moving average coefficients while $e_t$ is the forecasting error for the observation in time $t$. The reason we chose to test this forecasting method for our data is that it takes into account the strong auto-correlation that exists in our data. Moreover, ARIMA is a popular method used successfully with different types of datasets giving interesting results.

### 3.3.4. Holt-Winters Exponential Smoothing

Exponential smoothing (or ETS) is a forecasting method for uni-variate data where the prediction is a weighted linear sum of recent past observations, in which the weight assigned to past observations decreases exponentially. Therefore, the ETS model gives more importance to the most recent observations, i.e., the more recent are the observations, the higher is the associated weight. As our data contains trend and seasonality, we need to consider Holt-Winters Exponential Smoothing, which provides support for both of them as in the following equation.

$$\widehat{Y_{t+m}} = (S_t + mb_t) I_{t-L+m} \tag{4}$$

where:

- *y* is the observation in time *t*
- *S* is the smoothed observation
- *b* is the trend factor
- *I* is the seasonal index
- *L* is the number of periods is a season
- $\widehat{Y_{t+m}}$ is the forecast at *m* periods ahead

The reason we chose to test this forecasting method is that it performs well in those situations where there is a changing trend in the data and the last observations are the most important ones to consider.

### 3.3.5. Prophet

Prophet is a method developed by Facebook for forecasting time series data, based on an additive model. As described by its authors in [9], Prophet is based on a modular regression model with interpretable parameters that can be intuitively adjusted by the user. It was specifically designed for data that may contain non-linear trends and have weekly and daily seasonality, plus holiday effects. Given that our mobile phone dataset has precisely those features, Prophet seems to be a good candidate to test. Moreover, Prophet is robust to missing data and shifts in the trend, and can handle outliers quite well.

### *3.4. Deep Learning Neural Networks-Based Forecasting Methods*

In this subsection, we describe three different forecasting methods based on deep learning neural networks that we will use in our forecasting scenario. They are: a Multilayer Perceptron, a Convolutional and an Encoder-Decoder Long Short Term Memory neural network. The reason we are considering these methods is their ability to automatically learn the complex linear and non-linear relations that exist in the data such as the ones we show in Figure 4. In addition, they offer the possibility to perform one-step and multi-step forecasting given a data time window and several steps to predict. Finally, these methods can benefit from STL decomposition techniques as in [8] which make their forecasts results robust to large errors.

### 3.4.1. Multilayer Perceptron

A multilayer perceptron (MLP) is a neural network with a single hidden layer. Although with such a simple architecture, an MLP can learn any function, in particular by being able to approximate non-linear relationships between input and output variables. An MLP can perfectly learn from the series of past observations and use that knowledge to predict the next value or sequence of steps at test time. The training dataset for our MLP neural network is prepared as in Figure 7: in that example, the number of steps ahead to forecast is equal to 6, whereas the number of input timestamps is equal to 10. Clearly, by setting the input and output data it is possible to perform forecast with more steps in the future.

### 3.4.2. Convolutional Neural Networks

A Convolutional Neural Network (CNN) is a neural architecture that has been originally designed for the processing of images [10], but which has been further generalized to handle speech, signals, and text [11,12]. This model is based on the idea of receptive fields that are capable of automatically learning filters to extract features from raw data. Clearly, this capability of extracting features from the input can be exploited also in the scenario of time series forecasting [13]. As it happens with images, a CNN can process time series data such as those prepared in Figure 7, by learning one-dimensional receptive fields that can capture dependencies throughout the input signal, and by exploiting such information to perform forecasts.

### 3.4.3. Long Short-Term Memory Networks

Recurrent Neural Networks (RNN) are neural networks specifically designed to handle sequential data. For this reason, they make a perfectly suitable candidate approach for time series forecasting. There is however a specific type of RNN, named Long Short Term Memory (LSTM) network, which is particularly useful when there is a need to learn long-range dependencies. LSTM models have been successful in many applications, such as natural language processing, but they can be applied effectively to time series forecasting tasks as well. In addition to this, LSTM networks are capable of accurately modelling complex multivariate sequences [14]. For our forecasting model we chose an Encoder-Decoder LSTM model which accepts a sequence of uni-variate input data and produces as output a multi-step forecasting sequence.

### 3.4.4. Hybrid Cnn-Lstm Model

After testing with a CNN and an LSTM neural network it comes natural to combine and use them as a single model. We refer to this hybrid model as CNN-LSTM, and for our forecasting purpose we are using them together in an encoder-decoder architecture. We exploit the fact that a one-dimensional CNN is capable of reading the input sequence of the time series data and learn the the most important features. This way the CNN will be acting as the encoder inside the encoder-decoder architecture. The CNN architecture for the encoder has one convolutional layer. The output of such layer can then be interpreted by an LSTM decoder. The final output is handled by a deep neural network. More details on the architecture are given in the following subsection. A similar approach to the hybrid model described above is used in [15].

### *3.5. Tuning the Forecasting Methods*

As a pre-processing step, in all our experiments we aggregated the data so as to obtain one sample per hour instead of four as in the original dataset. We implemented the deep learning approaches with Keras (https://keras.io/) and the the statistical methods such as ARIMA, ETS and AR with the Statsmodels package (https://www.statsmodels.org/stable/index.html).

### 3.5.1. Statistical Methods Parameters

- ARIMA. To find the best set of parameters for the ARIMA model, we performed a grid-search by trying different combinations of values for its three parameters $a$, $d$ and $q$. The grid-search algorithms finds the best set of parameters for ARIMA by using as an optimization criteria the Root Mean Square Error of the residuals of the fitting process. Only the set of parameters yielding the lowest Root Mean Square Error was picked for performing forecast.
- Exponential Smoothing. As with what we did with ARIMA, also for ETS we tuned the parameters of the model by performing a grid-search with as criteria of optimization the RMSE error. Also in this case, only the set of parameters that yields the lowest RSME is picked, and this was done for every time series of each cell in the grid.
- Autoregression. For this forecasting model the Statsmodels library feature allows automatically selecting an appropriate lag value in order to make predictions.
- Prophet. For Prophet, instead, we did not optimize any parameter, but simply relied on the capability of its implementation to adapt to data as described by its creators in [9] .

### 3.5.2. Deep Learning Methods Hyper-Parameters

The architectural choices for all the neural network models have been designed as a trade-off between the possibility of representing more complex mapping functions (i.e., number of layers) and the opportunity of capturing the major possible number of mapping functions (i.e., number of nodes). Other than the network topology (number of layers and number of neurons/layers), we have

been considering, the dimension of the input data window, the number of epochs, the batch size, the optimizer, the loss function, and the learning rate.

We tested the input data window and the batch size which we set to 168 and 328 respectively. The reason we chose those two numbers is that the first contains the observations of one week while the second is a divisor of the training set. Through experimentation, we have seen that these two parameters give optimal results. Moreover, we monitored the training process in order to stop it at the point when performance on a validation dataset starts to degrade sensibly despite the tolerance parameter. We used the procedure of early stopping which is implemented within Keras API. Regarding the optimizer, we have been using Adam for MLP, CNN and LSTM while for the CNN-LSTM model we have been using SGD. To find the optimal learning rate for the CNN-LSTM model we use a technique that uses callbacks to tweak the learning rate using a learning rate scheduler so as to change the learning rates to a value based on the epoch number. By using this technique we are able to find the best learning rate which as shown in Figure 8a corresponds to $10^{-5}$. We use this learning rate and obtain the graphics in Figure 8b which shows that the training process is stable from a learning point of view with the loss decreasing progressively even for several epochs larger than 200.

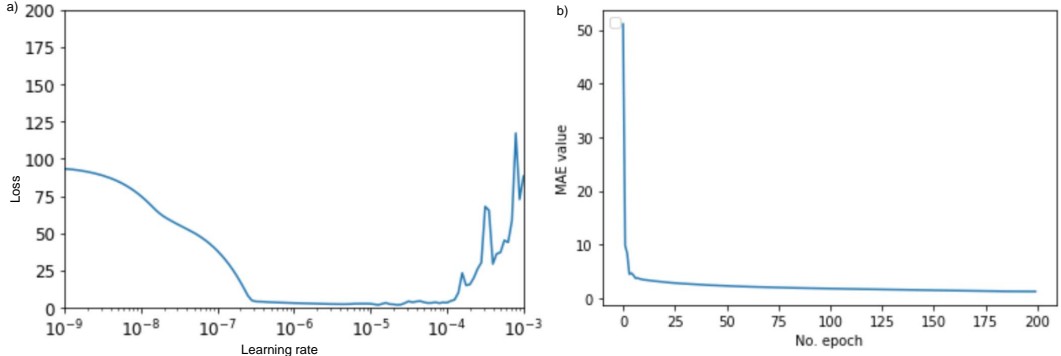

**Figure 8.** (**a**) The loss function of the CNN-LSTM with varying learning rate and (**b**) the loss function with the best learning rate chosen in the previous step which decreases up to 200 epochs.

- For the MLP neural network, we used a single hidden layer with 100 neurons. We analyzed the trend of the loss function with a different number of nodes from 100 to 900, observing no major differences. Regarding the number of hidden layers we empirically discovered that by adding more than one layer to MLP neural network we would just increase the time of training without benefits on the forecasting performance.

- For the CNN, we used a convolutional layer with 64 filters, a max-pooling with size equal to 2, followed by a dense layer with 50 neurons. Even in this case, we observed no significant changes with more complex architectures. As already said, the Adam optimizer was used during training.

- For the Encoder-Decoder LSTM, we used two layers containing 20 neurons each which we chose as a trade-off between simplicity and performance, similarly to what we did for MLP and CNN. As already said, the Adam optimizer was used during training.

- The CNN-LSTM architecture as shown in Figure 9 has one convolutional layer with 64 filters. Its output can then be interpreted by an LSTM decoder which contains 2 levels of 64 neurons each. Before the final output layer, two dense layers with 32 and 16 neurons, respectively, are used. For this model, we used the SGD optimizer and by experimenting with the learning rate as shown in Figure 8a we were able to pick the one that gives a stable decreasing loss as in Figure 8b. We have prepared a repository on Github with the CNN-LSTM (https://github.com/alketcecaj12/cnn-lstm) model with data from one cell.

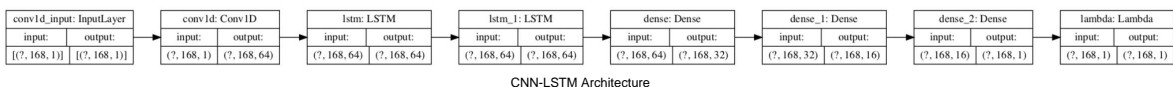

CNN-LSTM Architecture

**Figure 9.** The architecture of the CNN-LSTM hybrid model.

### 3.6. Model Evaluation and Forecasting Performance Measures

Model evaluation is crucial when comparing forecasting models, as its choice may affect the final results in relevant way. As customary, we use the walk-forward validation method for the evaluation of our statistical models, whereas for deep learning ones we use train-test validation. Moreover, for each member of the two categories of forecasting methods we measure the forecasting performance by using three different forecasting error measures in a multi-step forecasting scenario.

- Mean Absolute Percentage Error (MAPE), which is scale-independent forecasting measure and is defined as:

$$\text{MAPE} = 100 \cdot \sum_{t=1}^{N} \frac{|y_t - \hat{y}_t|}{y_t} \tag{5}$$

  where $y_t$ is the observation at time $t$ and $\hat{y}_t$ is the corresponding predicted value. In simpler terms, MAPE is the average error in percentage of the observation we are trying to forecast. MAPE is thus scale-independent, so it represents the perfect fit for our grid dataset composed of thousands of cells where each cell may have levels of mobile activity in a very different scale.

- Root Mean Squared Error (RMSE), a widely employed error measure defined as:

$$\text{RMSE} = \frac{1}{N} \sum_{t=1}^{N} (y_t - \hat{y}_t)^2 \tag{6}$$

- Maximum Absolute Error (MAXAE) considers the maximum error (in absolute value) that a given predictor can make, thus taking into account the worst-case scenario.

$$\text{MAXAE} = \max_{t} |y_t - \hat{y}_t| \tag{7}$$

There are of course other forecasting measures such as MAE (Mean Absolute Error) which is both popular and simpler to understand. However, in our case MAE would result misleading and this for two reasons. The first one is that there are "calm" periods in our data which are easy to predict but there are also periods which contain high variability. By using MAE we would not be able to show the distribution of the error in particular for high variability observations. This is also the reason we consider the MAXAE as an additional measure of performance which is particularly good in showing the maximum error of forecast. Finally, we compute each of these three forecasting measures for each one of the cells, thus we use a boxplot graphics to show their distribution along the cells.

### 4. Experiments

In this section, we illustrate the results of our experimental analysis aimed at comparing the forecasting performance of statistical and deep learning methods along different experimental directions and in particular under three forecasting scenario.

- We first consider a 24-h-forward forecasting scenario, which allows comparing the different methods by using the forecasting performance measures mentioned in Section 3.6.
- As a second experiment, we analyze the variation in the prediction error as a function of the number of steps to forecast. So in this case we show how varies each of forecasting measure by increasing the number of steps to forecast.
- Finally, we separately compare the considered approaches on two different types of cells in our dataset, namely those with regular and periodic time series, and those with high variability,

i.e., cells which contain strong anomalies and irregular trends. Two example time series, one for each cell category, are shown in Figure 10a,b respectively. We first define a criterion to distinguish between the two types of cells, and then evaluate the performance of deep learning and statistical methods for both types.

### 4.1. 24-h-Ahead Forecasting Analysis

In this subsection, we describe the forecasting performance of the statistical and deep learning methods for a multi-step prediction task. In particular, we consider 24-step-ahead forecasting, with a step equal to one hour. We test the forecasting methods illustrated above with each time series of each cell in our dataset. As the ours is a multi-step forecasting process, we compute the forecasting error by summing up the errors committed at the intermediate time steps. In this way, we obtain three types of forecasting errors (MAPE, RMSE and MAXAE) for each cell. Given that our dataset is a grid of cells and we compute the three errors for each cell, there will be a distribution of values for each of the three errors containing as many elements as the number of cells in the dataset. We represent this distribution with a boxplot as in Figure 11, which shows the distribution of RMSE for each forecasting method. More precisely, each boxplot contains as many elements as the number of cells, where each element is the RMSE value obtained for the time series of that cell with the corresponding method. Something we can immediately notice from this plot is that statistical methods typically perform better, with ARIMA being the best performing approach. Sometimes, even the naïve baseline model computing the seasonal average can outperform more sophisticated approaches such as CNN and LSTM, which typically require large datasets for training. The CNN-LSTM hybrid model is the best performing model among the machine learning methods, performing much better in comparison to statistical models, such as ETS and AR. The same conclusions can be drawn by looking at the MAPE performance measure, which is shown in Figure 12. However, something worth noticing in this graph is that for this kind of error, the performance of the CNN-LSTM model is quite similar to that of ARIMA.

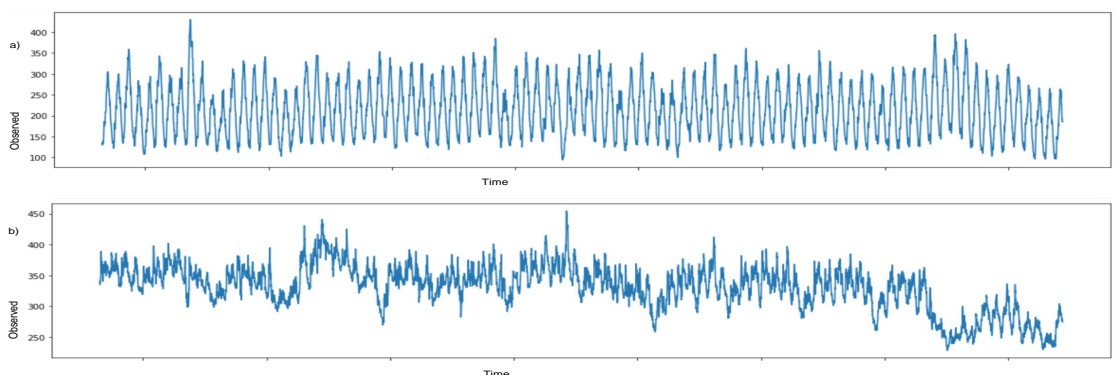

**Figure 10.** Two different types of cells with very different mobile activity patterns. The cell in (**a**) has a periodic, regular pattern, whereas the cell in (**b**) has an irregular pattern containing local and global trends with activity levels going up and down.

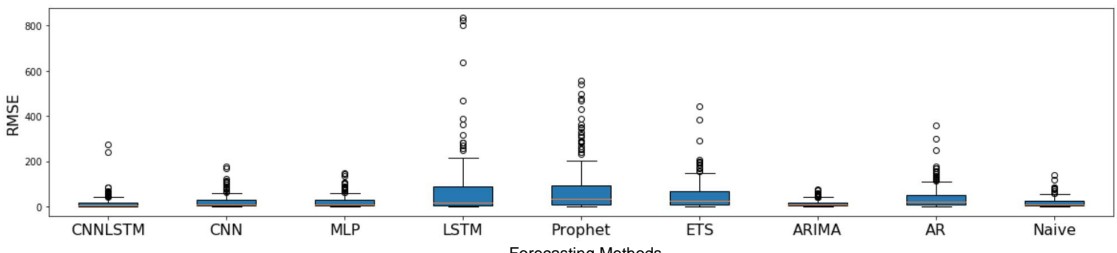

**Figure 11.** The distribution of RMSE across cells in a 24-h-ahead forecasting scenario.

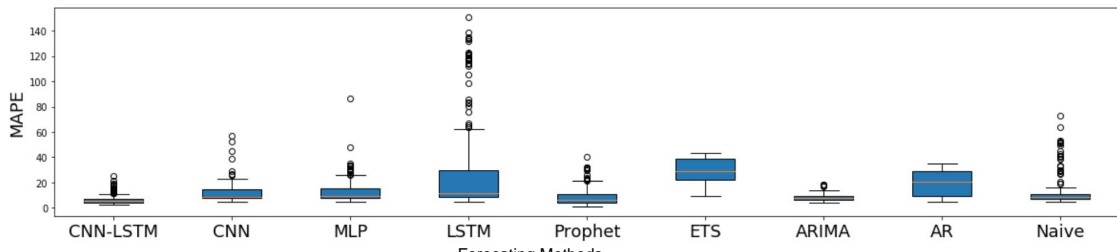

**Figure 12.** The distribution of MAPE across cells in a 24-h-ahead forecasting scenario.

When looking at MAXAE, represented in Figure 13, the gap between statistical and deep learning approaches is reduced, with the hybrid model CNN-LSTM performing better on average than ARIMA or any other statistical method. The MLP model also shows a better performance when compared to statistical methods such as ETS or AR. This behavior suggests that, in general, deep learning approaches are more robust to spikes and sudden changes in the time series. This point is important, because having an upper bound about the maximum error that we can make can give to the researcher the possibility to choose in which scenario to use a given forecasting method.

Table 1 summarizes all the results contained in the boxplot graphics presented above. By computing the mean and the standard deviation of each error distribution, we give an even more synthetic overview of the forecasting performance of each method. In particular, ARIMA and CNN-LSTM hybrid model show a similar performance for both MAPE and RMSE erros while for the MAXAE error the CNN-LSTM shows to perform better than ARIMA. More in general, deep learning methods outperform statistical methods when it comes to containing the maximum error.

**Table 1.** Mean and standard deviation for each of the three performance measures across all the considered cells sorted by MAPE error.

| Methods | MAPE | | RMSE | | MAXAE | |
|---|---|---|---|---|---|---|
| | Mean | Std.Dev | Mean | Std.Dev | Mean | Std.Dev |
| LSTM | 32.3 | 41.6 | 66.5 | 122.6 | 182.5 | 220.1 |
| ETS | 28.8 | 9.2 | 48.6 | 61.7 | 130.74 | 164.6 |
| AR | 19.7 | 10.3 | 36.9 | 49.6 | 116.3 | 149.8 |
| CNN | 12.0 | 9.1 | 20.5 | 26.7 | 68.8 | 90.88 |
| MLP | 11.6 | 7.4 | 21.5 | 27.4 | 72.5 | 93.1 |
| Naive | 9.0 | 2.4 | 17.7 | 20.7 | 91.8 | 132.5 |
| Prophet | 8.8 | 4.8 | 58.8 | 98.5 | 134.6 | 173.2 |
| CNN-LSTM | 7.9 | 3.7 | 14.9 | 28.6 | 32.6 | 54.79 |
| ARIMA | 7.7 | 3.9 | 13.9 | 15.9 | 57.0 | 69.9 |

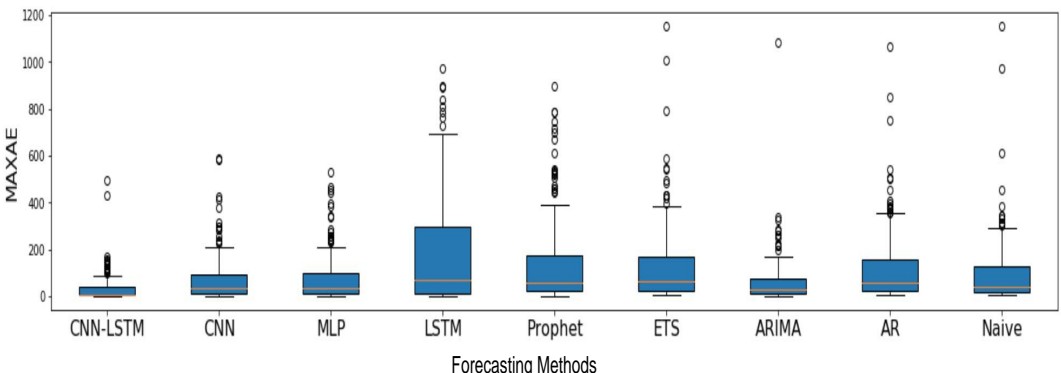

**Figure 13.** The distribution of MAXAE in a 24-h-ahead scenario.

### 4.2. Performance Along Different Forecasting Horizons

How does the forecasting error change when increasing the number of steps to forecast? Are there forecasting methods which are more robust than others to this process? In this subsection we give answers to these questions by exploring to what extent the forecasting errors vary along different horizons. Namely, we will make forecasts at 3, 6, 12, 24, and 48 steps forward. We omit in this case the naïve method which for how its nature works is used only for a 24 step forecasting scenario.

Clearly, when trying to predict long term dependencies by moving from a few hours up to two days (48 steps) in the future, we expect to make larger and larger errors. The plot in Figure 14 explains this concept pretty well. It has been generated while performing multi-step forecasting using ARIMA with walk-forward validation on a single cell time series. Each boxplot contains the distribution of the mean absolute error for the corresponding step of forecasting from one to 24 steps. The same plot for the case of MLP neural network is shown in Figure 15. Although we do not use the mean absolute error as a performance measure in our study, it come in handy in this case to illustrate the propagation of the forecasting error.

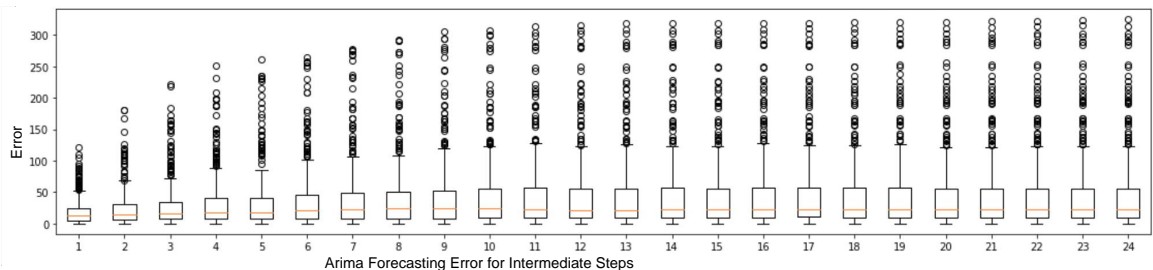

**Figure 14.** The distribution of absolute error for ARIMA in multi-step forecasting scenario. As it is natural to see, the absolute error increases with the number of steps forward to forecast.

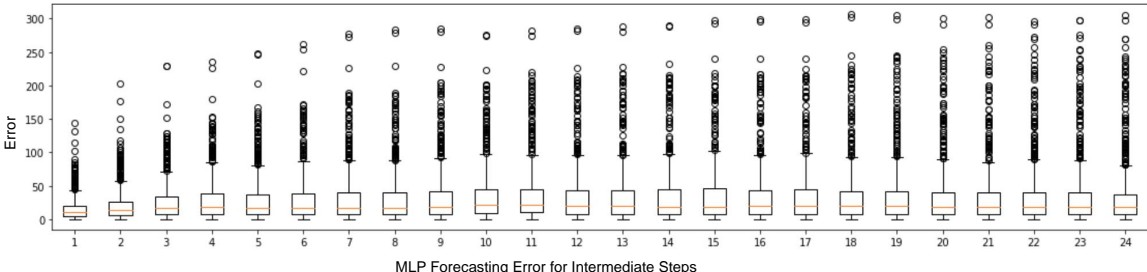

**Figure 15.** The distribution of absolute error for MLP neural network in multi-step forecasting scenario. The absolute error increases naturally with the number of steps forward to forecast.

In Figure 16 we show some first graphical results of this process regarding the three performance measures for ARIMA method. In the same way, Figure 17 shows the results for CNN-LSTM hybrid model.

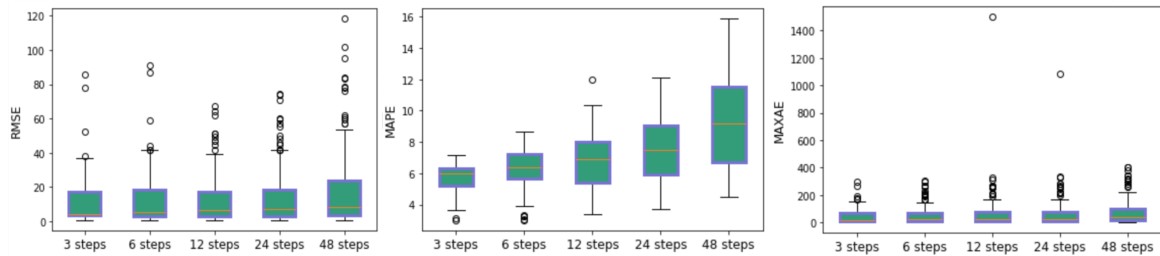

**Figure 16.** The distribution of RMSE, MAPE and MAXAE for ARIMA forecasting method in the five multi-step forecasting cases.

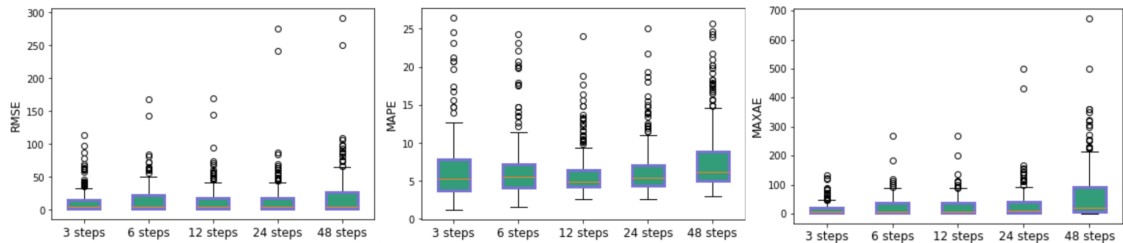

**Figure 17.** The graphics shows how is each type of error (RMSE, MAPE and MAXAE) distributed for the CNN-LSTM hybrid model in five different multi-step cases : 3, 6, 12, 24 and 48 steps.

Finally, in order to summarize the results for all the forecasting methods, we build three more tables as follows. In these tables we show the mean for each performance measure computed across all the cells of the grid. In general, for all the forecasting methods, there is a steady increase in the corresponding forecasting measure of performance when going from a few steps forecast to more steps in the future. Judging from the tables, there are two top-performing methods: the CNN-LSTM hybrid model and ARIMA method. They both show to perform well for MAPE and RMSE given in Tables 2 and 3 respectively. However, as Table 4 shows, the CNN-LSTM model outperforms ARIMA for MAXAE and does a better job in keeping this type of error low in the long term.

**Table 2.** The table shows how varies the mean of MAPE error across the cells for each method when transitioning from three steps forecasting up to 48 steps. Both statistical and deep learning methods show good results for 3–6 steps forecast. The deep learning methods such as CNN-LSTM show to be robust by keeping the MAPE low and almost constant in the long term.

| Methods | MAPE | | | | |
|---|---|---|---|---|---|
| | 3 Steps | 6 Steps | 12 Steps | 24 Steps | 48 Steps |
| LSTM. | 4.78 | 5.95 | 19.6 | 32.34 | 121.95 |
| ETS | 17.38 | 22.29 | 25.5 | 28.79 | 27.29 |
| AR | 4.29 | 8.26 | 16.5 | 19.72 | 21.34 |
| CNN | 6.15 | 8.37 | 10.22 | 11.99 | 14.23 |
| MLP | 6.35 | 7.68 | 9.48 | 11.63 | 14.06 |
| Prophet | 3.17 | 6.87 | 7.69 | 8.74 | 11.8 |
| ARIMA | 5.65 | 6.26 | 6.66 | 7.44 | 9.21 |
| CNN-LSTM | 6.45 | 6.41 | 5.93 | 6.35 | 7.8 |

**Table 3.** This table shows how varies the RMSE error from a three step forecasting case to six and up to 48 steps. For this type of error, the ARIMA method shows a slightly better performance than deep learning methods such as CNN-LSTM, in particular in the long run.

| Methods | RMSE | | | | |
|---|---|---|---|---|---|
| | 3 Steps | 6 Steps | 12 Steps | 24 Steps | 48 Steps |
| LSTM. | 10.0 | 13.09 | 21.24 | 66.46 | 313.35 |
| ETS | 41.23 | 55.08 | 45.01 | 48.59 | 47.47 |
| AR | 8.39 | 16.75 | 32.14 | 36.95 | 39.75 |
| CNN | 12.1 | 14.93 | 17.12 | 20.5 | 23.46 |
| MLP | 11.83 | 14.45 | 17.89 | 21.53 | 25.01 |
| Prophet | 31.5 | 31.48 | 37.0 | 78.84 | 57.95 |
| ARIMA | 11.38 | 12.25 | 12.57 | 13.87 | 17.94 |
| CNN-LSTM | 12.24 | 14.72 | 13.93 | 14.93 | 20.63 |

### 4.3. Low- and High-Variability Cells

It may be arguably easy to perform a multi-step forecast on a time series which keeps repeating periodically through time, without changing trends and with almost no variability in the data. Logically, even a naïve model which predicts the next sequence based only on the mean of past observations can perform fairly well. However, we know that this is not the case with mobile phone activity data where spikes and anomalies—mainly due to gatherings of people in a small space and frequent use of the mobile phone—are quite likely and in some cells even frequent. In this dynamic scenario, which is typical for a subset of cells in our dataset, the time series results difficult to predict.

**Table 4.** The table above summarizes the increase for the MAXAE error for every forecasting step from 3 and 6 up to 24 and 48. For this kind of performance measure, deep learning methods and in particular the CNN-LSTM outperforms the ARIMA method. More in general deep learning methods show a better forecasting performance than statistical methods.

| Methods | MAXAE | | | | |
|---|---|---|---|---|---|
| | 3 Steps | 6 Steps | 12 Steps | 24 Steps | 48 Steps |
| **LSTM.** | 43.78 | 54.57 | 78.57 | 182.51 | 868.55 |
| **ETS** | 123.17 | 156.53 | 126.46 | 130.74 | 132.73 |
| **AR** | 40.59 | 65.84 | 109.73 | 116.31 | 124.63 |
| **CNN** | 52.56 | 47.09 | 57.48 | 68.84 | 78.74 |
| **MLP** | 40.24 | 49.23 | 61.18 | 72.53 | 84.87 |
| **Prophet** | 39.23 | 52.52 | 63.57 | 134.56 | 122.25 |
| **ARIMA** | 46.11 | 51.16 | 63.15 | 62.54 | 74.16 |
| **CNN-LSTM** | 15.74 | 22.49 | 24.36 | 32.61 | 65.96 |

Therefore, we propose a further experiment for which we prepared two distinct datasets: a first one containing cells with low variability, as shown in Figure 10a, and a second one containing irregular, difficult to predict cells, with high variability, such as in Figure 10b. Hereafter we will refer to these two cases as low- and high-variability scenarios, respectively. To decide which cells belong to each category, we followed a method which is similar to the one used to describe the volatility of a financial asset return over time. In particular, for every cell we computed the percentage change $\Delta$ of the observation at time $t + 1$ with respect to the one at time $t$ as follows:

$$\Delta_y = 100 \cdot \frac{y_{t+1} - y_t}{y_t} \tag{8}$$

Thus, for each cell we compute the standard deviation of the percentage change $\Delta_y$ and then show the distribution across the cells, as in Figure 18. From this analysis, we chose to consider as cells with high variability those having a standard deviation of their percentage change higher than 15 in value. In the same way we chose as cells with low variability the ones having a standard deviation smaller than 9. These two thresholds allow us to partition the dataset into two subsets, each containing almost the same number of cells.

Then, we compute the three forecasting performance measures for every method, in a 24-steps forecasting use case for both low- and high-variability datasets. Figures 19 and 20 show the distribution of RMSE and MAPE respectively across all the cells in a low (left) and high (right) variability scenario. For these two performance measures, the top-performer methods are ARIMA and CNN-LSTM both in low and high variability context. As the graphics shows, they perform equally well though ARIMA is slightly better in particular regarding the RMSE performance measure.

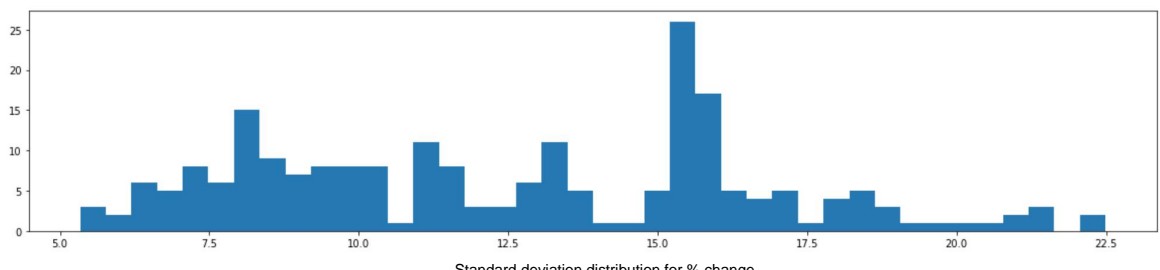

**Figure 18.** Distribution of the standard deviation of $\Delta_y$ across the cells. This distribution shows that there are cells with low variability and low variability in our grid dataset.

The MAXAE, whose distribution is shown in Figure 21, is where we can find a substantial difference between ARIMA and CNN-LSTM and more in general between statistical and deep learning methods. It is interesting to notice that the gap between ARIMA and deep learning approaches is much reduced (ARIMA showing even some outliers with very large errors), thus suggesting that deep learning approaches and in particular CNN-LSTM can successfully learn anomalous behaviors even in the case of high-variability time series cells.

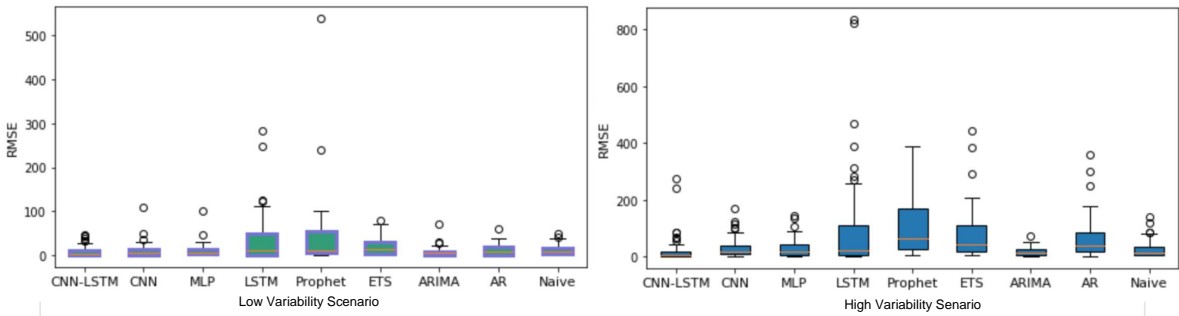

**Figure 19.** RMSE error distribution across the cells for both low (**left**) and high (**right**) variability scenario. Graphics shows that both the ARIMA method and CNN-LSTM deep learning method can give good results in high and low variability scenario.

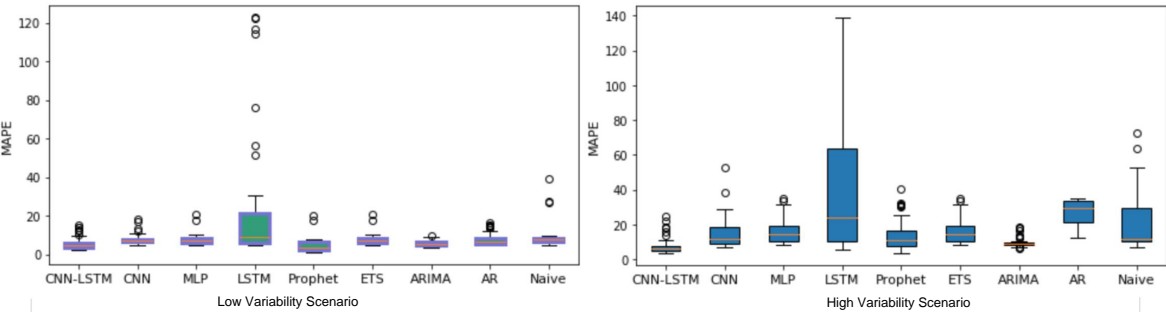

**Figure 20.** The boxplot graphics illustrates the distribution of MAPE across the cells and for each method. for both low and high variability scenario. Graphics shows how the ARIMA method can outperform deep learning methods for both high and low variability scenario.

Finally, based on the experiments we presented so far and trying to summarize the results in the clearest possible way, we present a list of conclusions as follows:

- Both statistical and deep learning methods show interesting results. In particular, the ARIMA statistical method and the CNN-LSTM deep learning model have proven to be the top-performers for each category.

- Performance measures such as RMSE and MAPE show that both ARIMA and CNN-LSTM are capable of delivering good forecasting results. These two measures show that none of such methods is significantly better than the other.
- The MAXAE measure shows that deep learning methods such as CNN-LSTM are significantly better than the statistical methods such as ARIMA in handling the anomalies in particular in high variability context and keeping the MAXAE low even for longer forecasting horizons.
- The ease of use of deep learning methods when compared to statistical methods makes them a better choice when the forecasting service has to be delivered for thousands of different time series as we do in this work. In this sense they provide a one-size-fits-all solution which as our work shows can yield interesting results.

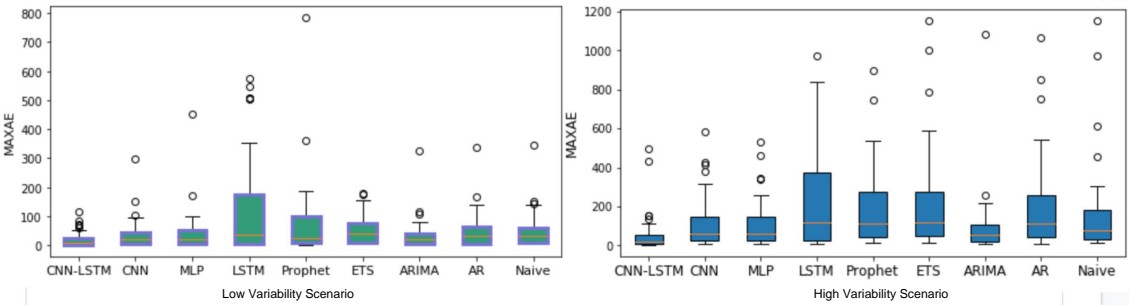

**Figure 21.** Each boxplot graphics indicates the distribution of MAXAE error across cells for a low and high variability scenario respectively in green and blue color. The graphics shows how deep learning methods can outperform statistical methods in a high variance scenario.

## 5. Related Work

Historically, the literature on time series forecasting has been dominated by linear forecasting methods such as ARIMA and its generalization for multivariate series VAR (or Vector Autoregression). The main reason for this fact is that those methods favor interpretability and are effective in many forecasting problems. Probably the most popular of these methods, ARIMA, has been used in a myriad of time series forecasting studies. For example, in [16] ARIMA has been used for performing forecast on international tourism time series data. The authors show that when finding a best-fitting ARIMA model, the forecasting error (RMSE) can be very low even for out-of-sample forecasts.

The ARIMA model is also applied in [17] for forecasting short-term electricity demand and thus network load by considering weather-network load relationship. In their paper, the authors demonstrate the effectiveness of the method, by comparing the results of the transfer function model and the univariate ARIMA model with conventional linear regression models. Another interesting work applies a combination of ARIMA with Fuzzy Expected Intervals methodology for fisheries management [18]. Finally the study in [19] applied ARIMA models to remote sensing data in order to forecast periods of drought in the Guanzhong Plain, China. According to authors, ARIMA models developed during this work, can be successfully used for the drought forecasting in the interested area.

Although successful in different scenarios, statistical methods such as the ones mentioned above depend too much on linear relationships for making predictions and this dependence may be a disadvantage. However, where statistical methods show limitations, deep learning approaches can be a valid alternative. This category of forecasting methods is relatively new but is quickly gaining popularity due to their ability to model complex nonlinear relationships and automatically learn arbitrary mappings from inputs to outputs. This wave of novel methods based on deep neural networks has sparked the debate inside the community of researchers about which category of methods performs best. The research in [20] was triggered by such debate. Although applying deep neural networks for time series forecasting is a relatively new trend, there are already thousands of works in the literature. Very recent work such as the one in [21] proposed an end-to-end deep learning-based

model to generate population flow from aggregated historical population variation data. The study in [22] shows how an ensemble of deep belief networks (DBN) can be successfully used for regression problems and time series forecasting. The study in [23] reports the results of the NN3 forecasting competition, highlighting the ability of neural networks to handle complex data, including short and seasonal time series. Finally, the work in [24] reviews the use of deep learning techniques for time-series analysis and forecasting. However, there is not only competition between statistical methods and deep learning ones but also a quest to integrate the two categories. For example, there have been attempts to combine neural networks with statistical methods such as ARIMA [25–27].

Finally, a third way to reason about time series forecasting is considering probabilistic models. Such models can define relationships between variables and thus they can be used to calculate probabilities on what will be the next observation. A probabilistic model that preserves known conditional dependence between variables are Bayesian networks, for which there exists a wide variety of works in the field of time series forecasting [28–32]. More similar to our setting where forecasting performance of statistical and machine learning methods are compared, is the study in [33], which compares artificial neural networks with traditional methods such Winters exponential smoothing, ARIMA model, and multivariate regression. The results in this work indicate that on average ANNs perform favorably with respect to the more traditional statistical methods, being able to capture the dynamic nonlinear trend and seasonal patterns, as well as the interactions between them. In the same way, though reaching a different conclusion, the study of [34] compares machine learning methods with statistical approaches with respect to the data size used for performing forecasts. This study shows that, when the data sample size grows, machine learning methods significantly improve their predictive performance. More recently the study in [5] shows that a hybrid neural network such as a Convolutional LSTM can perform significantly better than statistical methods such as ARIMA when making forecast on large scale. Concerning the works mentioned above, our study differs mainly on the following aspects listed below:

- as a major novelty, the type of data—aggregated mobile phone activity—being used is different from that of other works reviewed in this section, for what concerns both time granularity and spatial extension.
- the set of statistical and deep learning methods that are used for the comparative performance analysis as well as the performance measures used for such analysis.
- the experimental investigation of our research which does not aim to establish a winner between the forecasting methods but to contextualize which one is best in which context.

## 6. Conclusions

In this work, we compared the forecasting performance of statistical and deep learning methods on time series data from an aggregated mobile phone activity dataset. The results of our study show that both statistical and deep learning methods have advantages and limitations.

The main conclusions that we reach with this research work are that deep learning methods are easier to apply on time series forecasting tasks. In particular in tasks such as the one we explored in this paper in which the forecasting service should be provided for thousands of cells which may be very different one from the other in their time series. Besides simplicity, deep learning methods offer similar results to the most popular statistical methods such as ARIMA with the additional advantage of lowering the maximum error. Our experiments and in particular the MAXAE, have shown that deep learning methods are robust and avoid large errors mainly due to their ability to learn from data and also because of special data preparation techniques such as STL decomposition. Moreover, the deep learning methods applied in this study have shown to be more robust to parameter selection with respect to statistical approaches. They seem to be a one-size-fits-all solution which is the perfect match in a context such as ours. In addition to this, deep learning methods show to have greater benefits from the availability of larger data sets thus improving their forecasting performance when compared with statistical methods which typically do not improve their results when more data is available.

Statistical methods can instead give more precise results with lower forecasting errors on average but they are not the most suitable methods to use for our forecasting scenario. We reach this conclusion based on several findings and the most important ones are as follows. To use statistical methods successfully we need to have a very good knowledge of the data domain. Moreover, in our case we had to perform a careful analysis with time-consuming grid searching techniques for finding the optimal parameters of the statistical models such as ARIMA and ETS. On the contrary, deep learning methods do not require an expertise of data domain. They did not require us to go through any particular feature engineering process which may reveal be time-consuming and prone to errors due to arbitrary assumptions. Finally, given the strong linear relationship that exists in datasets such as aggregated mobile phone data, we reach the conclusion that statistical methods such as ARIMA are not able to perform significantly better than deep learning ones. This is particularly true when there are time-to-deliver constraints for the forecasting service and there is a need to minimize the maximum forecasting error. If these issues are a problem for the specific application then deep learning methods become a more time efficient and effective tool to use.

**Author Contributions:** Investigation, A.C.; Methodology, M.L.; Project administration, M.M. and F.Z. All authors have read and agreed to the published version of the manuscript.

**Funding:** Work supported by the POR-FESR 2014-2020 Project: POLIcy Support systEm for smart city data governancE-POLIS-EYE. PG/2018/631990. CUP E21F18000200007.

**Acknowledgments:** Work supported by the POR-FESR 2014–2020 Project: Policy Support System for smart city data governancE-POLIS-EYE. PG/2018/631990. CUP E21F18000200007.

**Conflicts of Interest:** There are no conflicts of interest between the authors and the funding parties.

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
