# Peer review of "Comparing Deep Learning and Statistical Methods in Forecasting Crowd Distribution from Aggregated Mobile Phone Data"

_applsci, doi:10.3390/app10186580_

Round 1
Reviewer 1 Report
This paper compares the performance of different models on predicting the density of mobile phone users. Particularly, this paper compares and discusses deep learning based models with traditional time series predictors.
I have following concerns on this paper:
- The technical contribution of this paper is limited. No new model or novel insight is provided in this paper.
- I believe that vanilla deep learning methods produce much worse results than conventional approaches. AR, ETS, ARIMA and prophet are designed for more general time series prediction, so it requires much less configurations and adaptations for this prediction task. However, deep learning is capable of modeling more complex spatial-temporal dependencies. Thus, a vanilla CNN or LSTM model is not suitable to represent a deep learning model.
- More justifications of the hyper-parameter settings (e.g. the hidden size, CNN filter size, how many layers are stacked) should be discussed and compared.
- More details of how CNN and LSTM are implemented in this work should be given. For example, how to model the long-term temporal dependencies in CNN and how to model spatial dependencies in LSTM?
Typos:
In the same section we illustrate how we we proceed...
Author Response
Reviewer 1 :
Point1
The technical contribution of this paper is limited. No new model or novel insight is provided in this paper.
Response1
Our work is about comparing statistical and deep learning forecasting methods on a mobile phone aggregated dataset. To the best of our knowledge, such an experimental comparison with a large set of forecasting methods/forecasting performance measures, and with this type of data has not been performed yet.
Point2
I believe that vanilla deep learning methods produce much worse results than conventional approaches. AR, ETS, ARIMA and prophet are designed for more general time series prediction, so it requires much less configurations and adaptations for this prediction task. However, deep learning is capable of modeling more complex spatial-temporal dependencies. Thus, a vanilla CNN or LSTM model is not suitable to represent a deep learning model.
Response2
While we agree that our MLP and CNN models are simple, the same cannot be said for the LSTM which is a bidirectional Encoder-Decoder LSTM. Moreover, we integrated our set of deep learning methods with a CNN-LSTM hybrid model which shows performance comparable with ARIMA on both short- and long-term forecasting problems.
Point3
More justifications of the hyper-parameter settings (e.g. the hidden size, CNN filter size, how many layers are stacked) should be discussed and compared.
Response3
For this point we give explanations in a special subsection in which we show how we tweaked the hyperparameters i.e. the training loss, the learning rate and the number of epochs of our model.
Point4
More details of how CNN and LSTM are implemented in this work should be given. For example, how to model the long-term temporal dependencies in CNN and how to model spatial dependencies in LSTM?
Response4
CNN have been widely employed in time-series forecasting applications. We refer to our section on related works, but we added more details in Section 3.4.2. While spatial dependencies in our dataset are an interesting issue which will be surely addressed in future work, in this work, we are not dealing with spatial relations between cells and their time-series.
Reviewer 2 Report
The authors propose two methods ( It. Deep learning-based and statistical methods) to forecast crowd distribution from aggregated mobile-phone data and come up with the following conclusions:
- Deep learning methods are preferable when it comes to simplicity and immediacy of use, are not time-consuming model selection, and as they are also aiming to reduce the maximum forecasting error.
- Statistical methods provide more precise forecasting results, but they require data domain knowledge and computationally expensive.
Comments:
- Are these conclusions based on your experiment? Since the conclusions are general features of both techniques across multiple domains. I couldn't see your specific conclusion.
- How could you reach the conclusion of computational expansiveness? Could you quantify/provide the number of training parameters of your methods?
- What makes your work different from others for example from the work of Zhang[5], since they also employed both deep learning-based and Statistical methods as you did in your work and even come up with a similar conclusion? Plus what is your motivation to use CNNs and LSTM separately? Rather it would be better if you trained them as a unified framework like that of [5].
- Page 12...you said “...to forecast (3-6 steps), LSTM and CNN are very competitive, and sometimes even superior in performance to ARIMA. Statistical approaches are instead typically more accurate for longer forecasting horizons.” why? Could you justify it?
As a final remark, it would be better if you provide the source of your dataset and again if you show the major architecture of your proposed model of both techniques.
Author Response
Reviewer 2:
Point1
Are these conclusions based on your experiment? Since the conclusions are general features of both techniques across multiple domains. I couldn't see your specific conclusion.
Response1
The conclusions are based on the experiments we did with every single cell time series in the grid dataset. About the conclusions, we rewrote both the small summary in 5 and the Conclusion section in 6 in order to be more clear and specific.
Point2
How could you reach the conclusion of computational expansiveness? Could you quantify/provide the number of training parameters of your methods?
Response2
On the computational expensiveness, we have shown the number of parameters of CNN-LSTM model and ARIMA. The number of parameters for our best performing CNN-LSTM model is 68,853. Being the computational cost strongly dependent on the dataset, we decided not to draw any general conclusion on computational expensiveness.
Point3
What makes your work different from others for example from the work of Zhang [5], since they also employed both deep learning-based and Statistical methods as you did in your work and even come up with a similar conclusion?
Response3
With respect to the work of Zhang [5], our research is different under a number of aspects as listed below:
1- The set of prediction methods used is different. We use 4 statistical methods while in Zhang[5] there is only one statistical method being used. About this, they use ARMA, while we use ARIMA. Though this might seem a small detail there is one parameter more to test in ARIMA which requires a longer curve fitting process in terms of time.
2- Performance measures. In Zhang [5], there is RMSE and MAE being used. Our work instead is using three performance measures RMSE, MAPE and MAXAE which help in evaluating better the performance of each forecasting method.
3- Type of data, time span and time granularity: in Zhang [5] they use a dataset spanning 16 days while ours is 122 days. The observations in our dataset are not about the individual mobile phone users as in Zhang [5] but instead our data is aggregated by using Elang measures. Thus we don’t need to use Kernel Density Estimation as in Zhang [5].
4- Approach of analysis. While in Zhang [5] there is a temporal and spatial analysis, in our work we focus only on the temporal one.
Point4
Plus what is your motivation to use CNNs and LSTM separately? Rather it would be better if you trained them as a unified framework like that of [5].
Response4
Thank you for this suggestion. We fixed this point by adding to our set of forecasting methods a hybrid CNN-LSTM model as required by the reviewers. We then reported its performance besides that of other DNN forecasting methods by updating all the related graphics and tables.
Point5
Page 12...you said “...to forecast (3-6 steps), LSTM and CNN are very competitive, and sometimes even superior in performance to ARIMA. Statistical approaches are instead typically more accurate for longer forecasting horizons.” why? Could you justify it?
Response5
We accept to modify this conclusion in particular after testing with CNN-LSTM we see that this model has a very similar performance to ARIMA. Moreover, standing to the performance measures, the CNN-LSTM seem to perform better than ARIMA for longer forecasting horizons. We rephrased the paper accordingly.
Point6
As a final remark, it would be better if you provide the source of your dataset and again if you show the major architecture of your proposed model of both techniques.
Response6
As stated in the paper, the source of our dataset is an Italian mobile phone operator. Following this suggestion, we added the graphical description of the architecture of the best-performing CNN-LSTM model (i.e., Figure 9 in the paper).